# Effect of APS Spraying Parameters on the Microstructure Formation of Fe_3_Al Intermetallics Coatings Using Mechanochemically Synthesized Nanocrystalline Fe-Al Powders

**DOI:** 10.3390/ma16041669

**Published:** 2023-02-16

**Authors:** Cezary Senderowski, Nataliia Vigilianska, Oleksii Burlachenko, Oleksandr Grishchenko, Anatolii Murashov, Sergiy Stepanyuk

**Affiliations:** 1Institute of Mechanics and Printing, Faculty of Mechanical and Industrial Engineering, Warsaw University of Technology, 02-524 Warsaw, Poland; 2Department of Protective Coatings, E.O. Paton Electric Welding Institute, 03680 Kiev, Ukraine; 3Department of Studies of Physical-Chemical Processes in Welding Arc, E.O. Paton Electric Welding Institute, 03680 Kiev, Ukraine

**Keywords:** plasma spraying, intermetallic coating, mechanochemical synthesis, conglomerate, splat test, deformation degree

## Abstract

The present paper presents a study of the behaviour of Fe_3_Al intermetallic powders particles based on 86Fe-14Al, 86Fe-14(Fe5Mg), and 60.8Fe-39.2(Ti37.5Al) compositions obtained by mechanochemical synthesis at successive stages of the plasma spraying process: during transfer in the volume of the gas stream and deformation at the moment of impact on the substrate. The effect of the change in current on the size of powder particles during their transfer through the high-temperature stream and the degree of particle deformation upon impact with the substrate was determined. It was found that during transfer through the plasma jet, there was an increase in the average size of sputtering products by two–three times compared to the initial effects of mechanochemical synthesis due to the coagulation of some particles. In this case, an increase in current from 400 to 500 A led to a growth in average particle size by 14–47% due to the partial evaporation of fine particles with an increase in their heating degree. An increase in current also led to a 5–10% growth in particle deformation degree upon impact on the substrate due to the rising temperature and velocity of the plasma jet. Based on the research, the parameters of plasma spraying of mechanically synthesized Fe_3_Al intermetallic-based powders were determined, at which dense coatings with a thin-lamellar structure were formed.

## 1. Introduction

Iron aluminide-based intermetallic alloys, due to their physicochemical and mechanical properties, as well as their stable structure and resistance to high-temperature corrosion, are promising materials for heat exchangers, nuclear reactor components, automotive exhaust systems, etc. [1,2,3,4,5]. The main advantage of iron aluminides over heat-resistant nickel alloys and stainless steels is the availability and low cost of the base iron component, as well as their ease of processing.

Iron aluminides have found a wide practical application as protective coatings obtained by thermal spraying, especially HVOF, plasma (PS), and Arc and D-gun spraying [2,5,6,7,8,9,10,11]. Unlike in D-gun spraying [10], during the PS process, powder particles falling into the high-temperature plasma jet (PJ) are melted and transferred to the substrate as droplets. In high-temperature flight, phenomena such as dispersion, coagulation, and changes in microstructure and phase composition occur in particles [12,13]. When the molten drop impacts the surface of the substrate, it spreads, solidifies, and forms a surface layer in the form of a splat. The structure and properties of coatings depend on the powder particles’ state during PS spraying. Completely molten particles contribute to the formation of dense layers and lead to reduced porosity. The degree of particle melting depends on the characteristics of the PJ (velocity and temperature, viscosity and thermal conductivity of the gas environment, degree of dissociation and ionisation of the gas molecules) and the material properties of the atomised particles (density, heat capacity, thermal conductivity, heat of fusion) [14].

At present, one of the ways to obtain composite intermetallic powders of the Fe-Al system for thermal spraying is the mechanochemical synthesis method (MCS) [15,16,17].

It should be noted, however, that unlike the D-gun process [9,10], the literature needs a comprehensive approach to studying the formation of plasma coatings using intermetallic powders of the Fe-Al system, including those obtained by the MCS method. Therefore, this study aimed to investigate the effect of changing the current intensity on the physicochemical processes occurring during forming Fe_3_Al-based intermetallic coatings by plasma spraying.

## 2. Materials and Methods

Iron aluminide powders obtained by mechanochemical synthesis were used as research material, while iron, aluminium, aluminium alloy (Al5Mg), and Ti37.5Al intermetallic powders were applied as starting materials. Using Ti as an alloying element allows for realising several mechanisms of iron intermetallic strengthening, namely structure ordering, strengthening with dispersed inclusions, and the formation of coherent microstructures. Titanium differs by significant solubility in a solid state in Fe–Al phases that result in Fe_3_Al structure stabilising around the FeAl structure at high temperatures. Strengthening with dispersed precipitations of hexagonal Laves phase (Fe, Al)_2_Ti can take place in addition to strengthening due to structure ordering in the Fe–Al–Ti system. Furthermore, there is a specific range of composition in the Fe–Al–Ti system, where coherent structures [18] are formed. If an element such as Mg is used for alloying powders based on the Fe_3_Al intermetallic compound, it is possible to expect strengthening by incoherent compounds. Commercially available powders of the alloys Al5Mg and Ti37.5Al were used instead of the pure elements of aluminium, magnesium, and titanium to reduce the degree of oxidation of the MCS products.

The MCS process was performed in a planetary-type mill «Aktivator 2SL» for 5 h. The relation of the mass of balls to the mass of powder was 10:1. The central axis of the mill triboreactor was rotated at a 100 rpm rate; drums rotated around their axis at a 1500 rpm rate. Parts of the jar and milling agents were manufactured of 100Cr6 steel. The MCS process was performed in the air. Surface-active substances (SAS), namely oleic acid, were added to the mixture to prevent pickup of processed charge on the milling agents and jar wall, and to intensify the process of new synthesis phases.

The amount of aluminium alloy powder introduced in the mixture with iron powder was selected for formation in MCS of Fe_3_(Al, Mg) intermetallics in the case of AlMg that corresponded to 14 wt.% of Al-alloy and (Fe, Ti)_3_Al in the case of TiAl intermetallics. In the latter, the variant amount of introduced TiAl was 39.2 wt.%.

The microstructure, chemical, and phase composition of the MCS powder products (as an intermetallic compound based on Fe_3_Al iron aluminide) were confirmed, respectively, by SEM/EDS and XRD analysis using a JEOL 5310 microscope (Japan) operating at 20 kV, and a XRD Simens D-500 diffractometer (Germany) with CoKα radiation (λ= 0.178897 nm). An angular step size of 0.02°/min and a step time of 5 s per point were used, respectively.

The appearance and X-ray patterns of MCS powders of Fe_3_Al, Fe-AlMg, and Fe-TiAl systems are shown in Figure 1 and Figure 2. The chemical composition of MCS powders is presented in Table 1.

The characteristics of the MCS powders used in work for investigating the coating formation process during plasma spraying are shown in Table 2 [15,16].

The crystallite size was estimated using the Scherrer equation:D = kλ/β cosθ
where k is the Scherrer constant (≈0.94), λ is the wavelength of the radiation used (for Co λ = 1.78897 Ǻ), θ is the reflection angle, β is the true broadening of the X-ray line.

Based on the crystallite size calculations, it can be noted that the powders obtained by the MCS method are nanocrystalline.

The absence of fluidity in MCS products is related to the high specific surface area of the particles, and the challenge is to uniformly feed these powders into the stream during PS [19]. To uniformly feed MCS products into PJ, they were conglomerated with a 5% polyvinyl alcohol solution in water, dried, and sifted for 40–80 µm particles. 

The choice of PS modes for Fe_3_Al–based intermetallic materials was carried out using the CASPSP software version 3.1 [20]. This software is designed for the computer simulation of turbulent plasma jets used in coating spraying and for modelling the heating and transport of atomised particles in such jets. Based on the analysis of the heating and transport of Fe_3_Al particles in the range of 40–80 μm in PJ, it was found that, in terms of total melting and the lack of evaporation of the particles, the most reasonable parameters are a current of 400–500 A and a plasma gas (PG) Ar+N_2_ flow rate of 25 SLPM (Figure 3). The use of other modes for spraying Fe_3_Al-based powders is irrational due to the absence of the complete melting of particles at a current of less than 400 A (Figure 4a) and the possibility of the material evaporation at a current of more than 500 A (Figure 4b).

Powder spraying was carried out using the UPU–8M atmospheric plasma spraying device at different arc current (I) parameters. Changing the current significantly affects the temperature and velocity of PJ, which determines the heating and velocity of the particles during transport in the stream [21]. The voltage, plasma gas (mixture Ar+N_2_) flow rate, and powder feed rate were constant in all experiments. The Ar/N_2_ ratio (7.3/1) and the flow rate of the plasma gas mixture (25 SLPM) made it possible to ensure the stable operation of the plasmatron at a voltage of 40 V. The modes of PS operation are presented in Table 3.

Studies of the PS processes occurring in the powder during spraying were carried out by collecting particles transferred through the plasma jet into the water at a distance of 120 mm from the end of the plasma torch to the water’s surface. The analysis of the size and microstructure of the particles of sprayed powders collected in the water bath will determine the processes taking place with the powder particles in a plasma jet, on which the further formation of the coating structure depends.

The state of the sprayed particle material after impact on the substrate was studied using the splat test. Spraying was carried out by moving polished stainless steel plates of 50 × 30 × 0.5 mm in a plane perpendicular to the jet axis (Figure 5). As a result, single particles of sprayed material deformed upon contact with the substrate surface (splats) were fixed on the samples.

To investigate powders sprayed into the water and evaluate the degree of deformation of particles after their impact on the substrate, a complex technique was used, including the analysis of particle size distribution–ASOD-300 laser analyser (Novatek-Electro, Odesa, Ukraine); metallographic examinations–optical microscope “Neofot-32” with an attachment for digital photography; measurement of splats diameters (D) and classification of splats by appearance–optical microscope Jenavert; scanning electron microscopy (SEM)–scanning electron microscope JSM-6390LV (JEOL, Warsaw, Poland) with an attachment for energy dispersive analysis INCA in the secondary electron mode, in low vacuum (10^–4^ Pa), with an accelerating voltage of 20 kV.

Chemical etching of metallographic cross-sections was used to reveal the structure of powders sprayed into water. For Fe_3_Al and Fe-AlMg powders, a 10% alcoholic solution of nitric acid was used for 4–5 min; the Fe-TiAl powder was etched with a solution of HF+HCl+HNO_3_+water for 2–3 min. Etching was carried out for the powders sprayed in Mode 3.

The microhardness of the powders and coatings was determined in a microhardness tester PMT-3. To quantitatively analyse the content of pores in the coatings, an optical technique (image analysis method) was used, which consists of determining the area per detected pores relative to the entire area of the coating cross-section (ASTM E2109-01). Image-Pro Plus 7 software processed the digital image of the microstructure of the coatings.

## 3. Results and Discussion

The appearance of powders of Fe_3_Al, Fe-AlMg, and Fe-TiAl systems sprayed into the water at different currents is shown in Figure 6, Figure 7 and Figure 8. Analysis of the appearance of the powders showed that most of the particles (~95%) are spherical, indicating their complete melting during transferring through the plasma jet.

It is known [22] that when a particle containing aluminium enters the oxygen-containing zones in a plasma stream, the process of aluminium oxidation develops with the appearance of an aluminium oxide film on the particle surface. By analysing the chemical composition of the powders, the presence of oxygen on the particle surfaces can be noted (Table 4, Table 5 and Table 6), which indicates the formation of oxides (in particular, aluminium oxide–Figure 6c, Table 4, Spectrum 1; Figure 7b, Table 5, Spectrum 1). 

The oxygen content in powders of the Fe-TiAl system is, on average, two times higher than the oxygen content in Fe_3_Al and Fe-AlMg powders. This is explained by the tendency of Ti and Al powder components to oxidize and their higher content in the initial powder mixtures. The reduced iron content in powders of the Fe-TiAl system also indicates the formation of an oxide film on the particle surface.

The microstructure of powders (Figure 9, Figure 10 and Figure 11) shows that oxides on the surface of the particles are arranged in the form of thin films or obtain a domed shape as a result of the movement of particles in the turbulent plasma stream (indicated by arrows at Figure 9, Figure 10 and Figure 11c).

Histograms of the particle size distribution of Fe_3_Al, Fe-AlMg, and Fe-TiAl powders after the conglomerates transferred through the plasma stream at different currents are shown in Figure 12, Figure 13 and Figure 14. It can be seen that most of the powder particles (>55%) are in the size range of 10…30 µm. The reduction in particle size compared to the conglomerates fed into the plasma stream is related to the binder burnout during the interaction of the high-temperature plasma stream with the conglomerates.

MCS products also simultaneously fuse with the destruction of conglomerates in the plasma stream. As a result, the average particle size increases 2–3 times relative to the initial MCS products (Table 7).

The increase in current leads to a 38–57% decrease in the number of particles within the size range of 10...20 μm and a 14–47% increase in the average particle size (Figure 15). This may be due to the partial evaporation of small particles with increased heating.

Analysis of the shape of individual splats on substrates is one of the factors in optimizing plasma spraying methods. The degree of particle deformation D/d on impact with the sprayed surface determines the particle’s surface contact area. The larger deformed particle size D compared to the initial particle size d in the gas stream in front of the sprayed surface, the greater probability of strong adhesion of the contacting materials, with all other conditions being equal.

The appearance of splashes obtained by spraying Fe_3_Al, Fe-AlMg, and Fe-TiAl powders as a function of current intensity is shown in Figure 16, Figure 17 and Figure 18. 

From the splats obtained during the interaction of powder particles with the substrate in all three modes, it can be concluded that the particles are in a fully molten state at the moment of impact with the substrate and have a disk shape. After the impact of the particles on the substrate and spread across the surface, the central part of the splats turns out to be unfilled by the material. This is explained by the fact that, inside the droplet, cavitation processes occur when it hits the surface of a solid; i.e., bubbles form and grow as the pressure drops to the saturation vapour pressure. The bubbles break through the liquid coating of the droplet and form crater-like holes in the deformed powder particle [23].

The results of estimating the degree of deformation of D/d particles after impact on the ground are shown in Table 8.

The dependence of the average splat diameter and the degree of particle deformation on the current is shown in Figure 19. As can be seen, as the current increases, the average diameter of the splats and the degree of deformation of the particles also increase. This is due to the increase in the temperature and PJ velocity, which leads to a decrease in the surface tension and viscosity of the molten particle, and an increase in the impulse and pressure that act on the particle when it impacts the substrate.

As a result of the plasma spraying of these powders, the coating layers are formed from completely melted and deformed particles, so the coatings have a dense thin-lamellar structure with a small number of oxide films at the lamella boundaries (Figure 20, Figure 21 and Figure 22). The coatings have dense adherence to the steel substrate, and no delamination defects are observed.

During the spraying of coatings of intermetallic powders, their phase composition does not completely coincide with the composition of the initial powders, which is associated with the active development of the process of particle oxidation during their flight. Oxides are present in all coatings. In the Fe_3_Al coating, Al_2_O_3_ aluminium oxide is present; in the Fe-AlMg coating, complex oxide MgAl_2_O_4_ and in the Fe-TiAl coating, titanium oxide TiO are present (Figure 23). In the Fe-AlMg coating, as in the original MCS powder, a solid solution of Mg in the Fe_3_Al intermetallic compound is retained. In the Fe_3_Al coating, the FeAl intermetallic phase is noted in addition to the base phase. During the spraying of the Fe-TiAl powder, the solid solution (Fe_1−x_TiAl_x_) was transformed into the intermetallic phase (Fe, Ti)_3_Al.

Changes in the phase composition of the coatings do not occur when the spraying mode changes.

The characteristics of coatings obtained under different modes of plasma spraying are given in Table 9.

As expected, the porosity of the coatings decreases (on average by 2%) with an increase in current from 400 to 500 A (Figure 24a), which is associated with a higher kinetic energy of particles upon impact with the substrate and a greater degree of their deformation with increasing current.

The results of measuring the microhardness of Fe_3_Al and Fe-AlMg coatings show a slight increase in microhardness relative to the initial MCS powders. This is most likely due to the formation of oxides in the coatings during spraying. In the case of sputtering of the Fe-TiAl powder, an increase in the microhardness of the coating relative to the microhardness of the initial MCS powder by ~2.3 times is noted. This is due to the formation of an intermetallic phase (Fe, Ti)_3_Al in the coating with additional hardening of the coating by the TiO oxide phase.

From Figure 24b, the increasing current slightly increases the hardness values for all types of coatings related to the content of pores in the coatings. Increasing currents decrease the coatings’ porosity, which lead to an increase in microhardness values.

Thus, to obtain plasma coatings with a dense lamellar structure and high adhesion and cohesion strength from Fe_3_Al-based iron aluminide powders obtained by the MCS method, it is advisable to carry out the spraying process at the following spraying parameters: voltage of 40 V, current of 500 A, plasma–Ar/N_2_ gas mixture in the ratio of 7.3:1.

## 4. Conclusions

The behaviour of particles of iron aluminides powders (Fe_3_Al, Fe-AlMg, and Fe-TiAl) obtained by mechanochemical synthesis was analysed at the stage of their transfer through the volume of the high-temperature plasma jet and after impact on the sprayed surface, depending on the intensity of the arc current in the plasma spraying process;It was established that, as a result of the contact of molten particles with oxygen-containing zones of the plasma jet, aluminium oxide films are formed on the surface of the particles, which acquire a dome shape when the particles are carried by the turbulent plasma jet;It was noted that, during the transfer of particles through the plasma jet, the initial particles are coagulated, resulting in their average size increasing from 9–15 µm to 19–31 µm;An increase in current leads to a 5–10% increase in the degree of particle deformation upon collision with the substrate as a result of the increase in temperature and plasma jet velocity;It has been found that an increase in current from 400 to 500 A during the plasma spraying of powders based on the Fe_3_Al intermetallic compound leads to the formation of coatings with a denser structure due to an increase in the degree of particle deformation during the formation of the coating layer;As a result of the research carried out, it was determined that, for the formation of coatings with a thin-lamellar dense structure and a dense boundary with a steel substrate from MCS intermetallic Fe_3_Al-based powders, it is advisable to use the following plasma spraying parameters: voltage 40 V, current 500 A, plasma-forming gas–Ar/N_2_ mixture at a ratio of 7.3:1.

## Figures and Tables

**Figure 1 materials-16-01669-f001:**
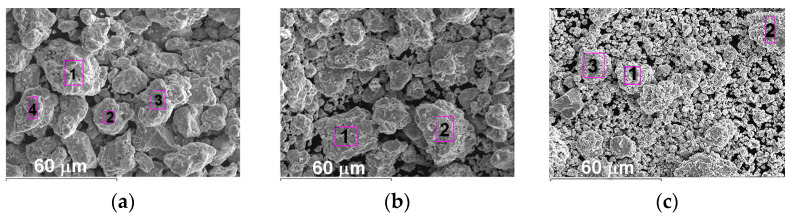
Appearance MCS powders: (**a**) Fe_3_Al, (**b**) Fe-AlMg, (**c**) Fe-TiAl.

**Figure 2 materials-16-01669-f002:**
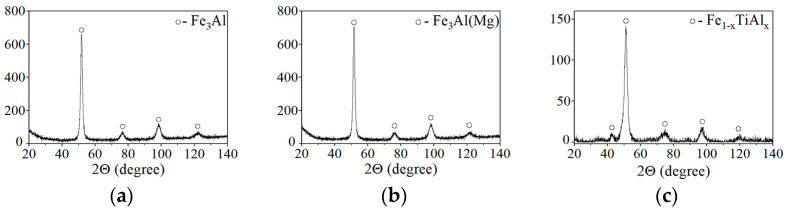
X-ray patterns of MCS powders: (**a**) Fe_3_Al, (**b**) Fe-AlMg, (**c**) Fe-TiAl.

**Figure 3 materials-16-01669-f003:**
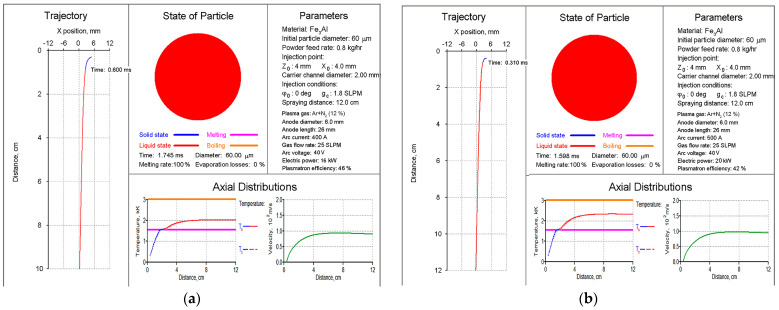
Graphical result of simulation by the CASPSP software of plasma spraying of Fe_3_Al particles at the current of 400 (**a**) and 500 A (**b**).

**Figure 4 materials-16-01669-f004:**
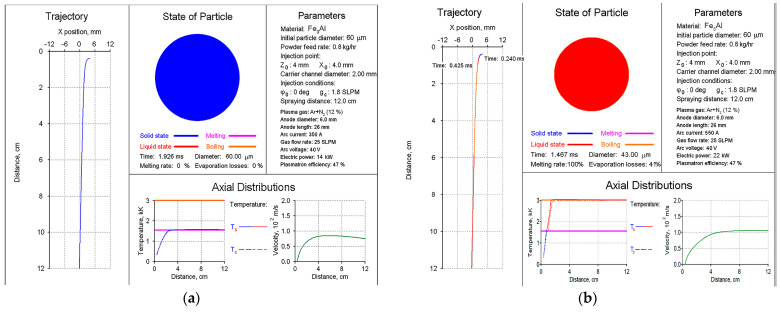
Graphical result of simulation by the CASPSP software of plasma spraying of Fe_3_Al particles at the current of 350 (**a**) and 550 A (**b**).

**Figure 5 materials-16-01669-f005:**
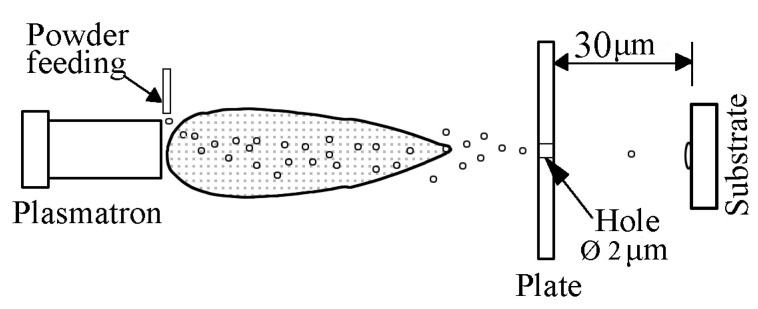
Scheme of the splat test.

**Figure 6 materials-16-01669-f006:**
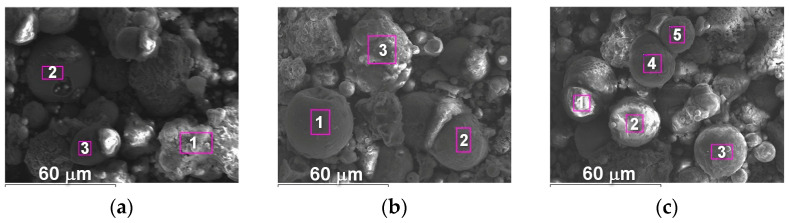
The appearance of Fe_3_Al powders after transfer through the plasma jet: (**a**) I = 400 A, (**b**) I = 450 A, (**c**) I = 500 A.

**Figure 7 materials-16-01669-f007:**
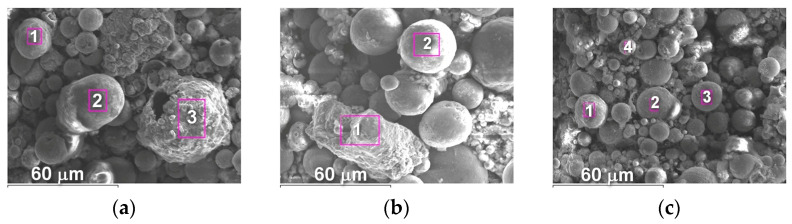
The appearance of Fe-AlMg powders after transfer through the plasma jet: (**a**) I = 400 A, (**b**) I = 450 A, (**c**) I = 500 A.

**Figure 8 materials-16-01669-f008:**
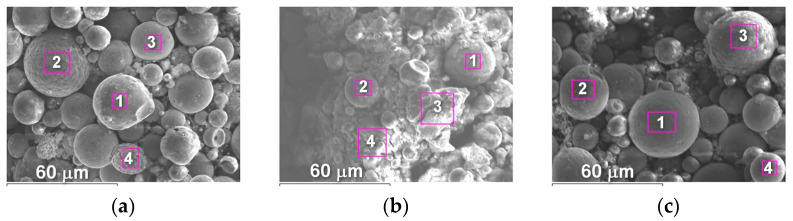
The appearance of Fe-TiAl powders after transfer through the plasma jet: (**a**) I = 400 A, (**b**) I = 450 A, (**c**) I = 500 A.

**Figure 9 materials-16-01669-f009:**
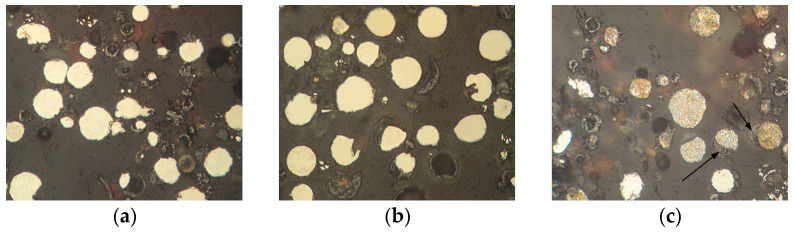
Microstructure (×400) of Fe_3_Al powders after transfer through the plasma jet: (**a**) I = 400 A, (**b**) I = 450 A, (**c**) I = 500 A (etched).

**Figure 10 materials-16-01669-f010:**
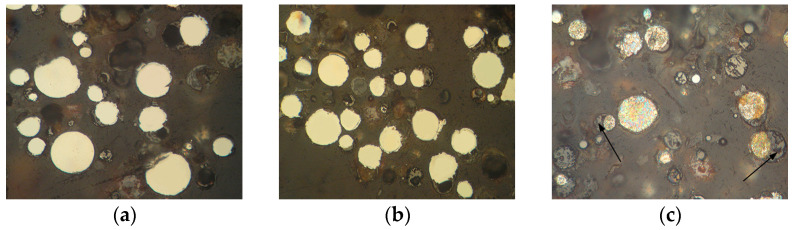
Microstructure (×400) of Fe-AlMg powders after transfer through the plasma jet: (**a**) I = 400 A, (**b**) I = 450 A, (**c**) I = 500 A (etched).

**Figure 11 materials-16-01669-f011:**
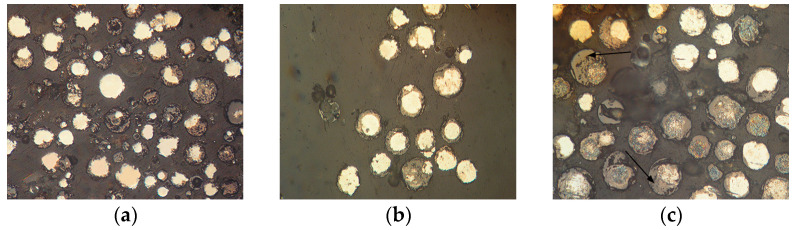
Microstructure (×400) of Fe-TiAl powders after transfer through the plasma jet: (**a**) I = 400 A, (**b**) I = 450 A, (**c**) I = 500 A (etched).

**Figure 12 materials-16-01669-f012:**
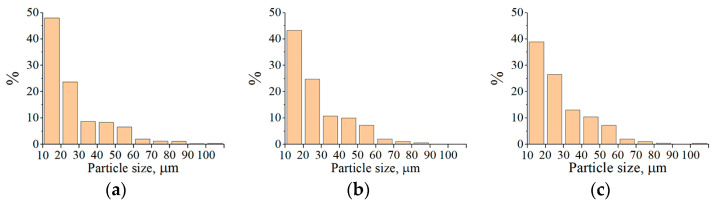
Histograms of the particle size distribution of Fe_3_Al powders sprayed into the water: (**a**) I = 400 A, (**b**) I = 450 A, (**c**) I = 500 A.

**Figure 13 materials-16-01669-f013:**
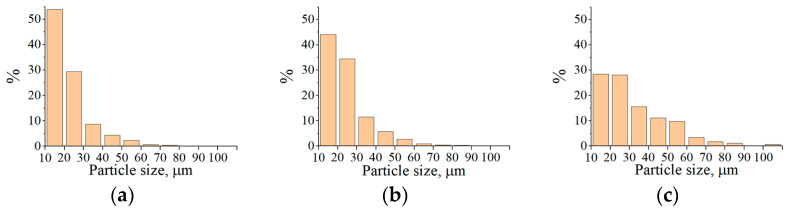
Histograms of the particle size distribution of Fe-AlMg powders sprayed into the water: (**a**) I = 400 A, (**b**) I = 450 A, (**c**) I = 500 A.

**Figure 14 materials-16-01669-f014:**
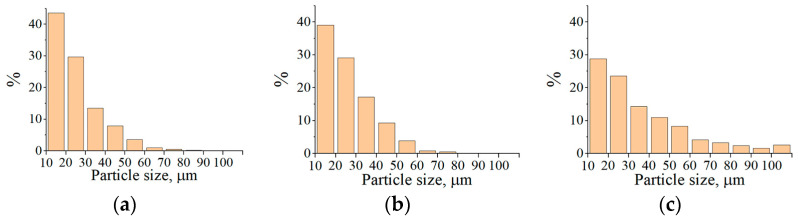
Histograms of the particle size distribution of Fe-TiAl powders sprayed into the water: (**a**) I = 400 A, (**b**) I = 450 A, (**c**) I = 500 A.

**Figure 15 materials-16-01669-f015:**
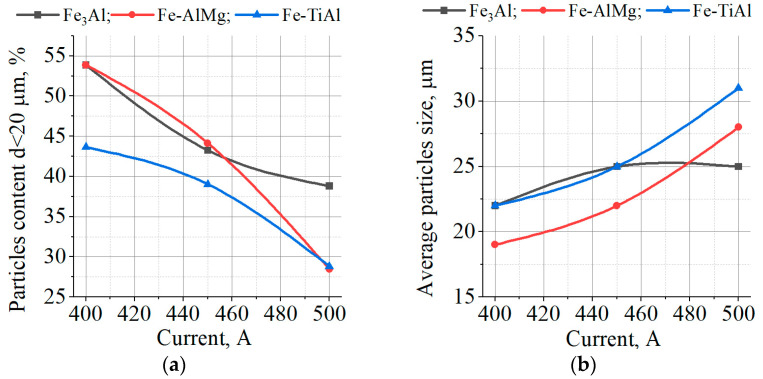
Effect of current change: (**a**) on particle content of <20 µm; (**b**) on the average particle size of powders.

**Figure 16 materials-16-01669-f016:**
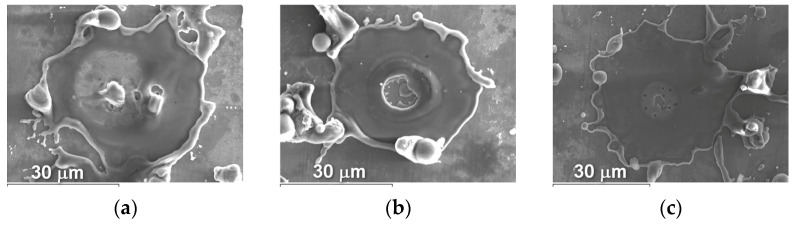
Splashes of Fe_3_Al powders after impact with steel substrate at (**a**) I = 400 A; (**b**) I = 450 A; (**c**) I = 500 A.

**Figure 17 materials-16-01669-f017:**
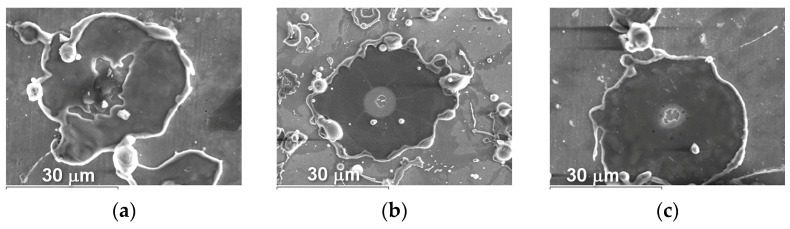
Splashes of Fe-AlMg powders after impact with steel substrate at (**a**) I = 400 A; (**b**) I = 450 A; (**c**) I = 500 A.

**Figure 18 materials-16-01669-f018:**
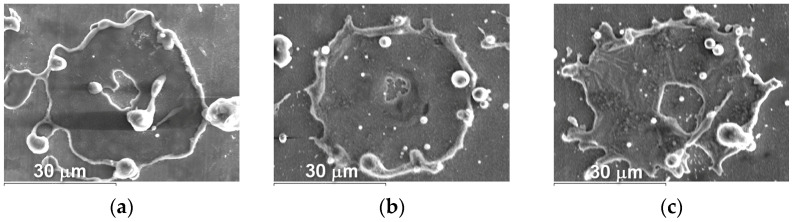
Splashes of Fe-TiAl powders after impact with steel substrate at (**a**) I = 400 A, (**b**) I = 450 A, (**c**) I = 500 A.

**Figure 19 materials-16-01669-f019:**
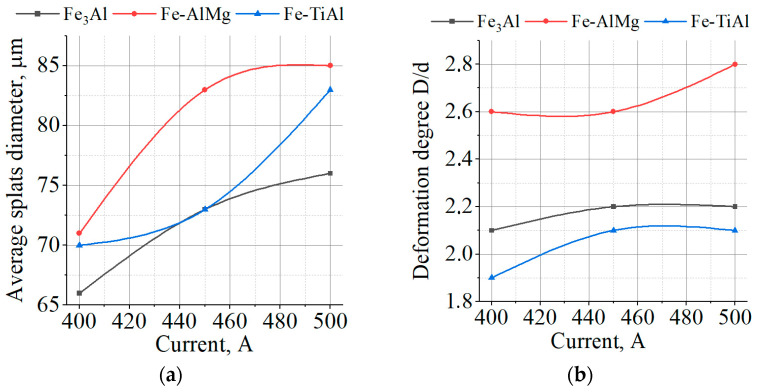
Effect of current change on (**a**) average splats’ diameter; (**b**) particles’ deformation degree.

**Figure 20 materials-16-01669-f020:**
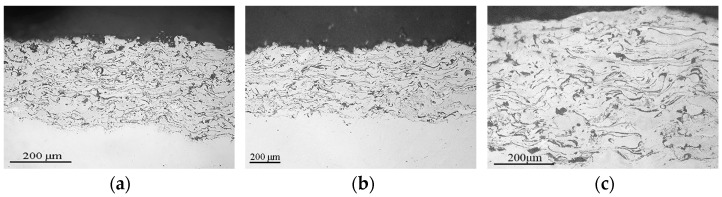
Microstructure of Fe_3_Al plasma coatings sprayed at: (**a**) I = 500 A; (**b**) I = 450 A; (**c**) I = 500 A.

**Figure 21 materials-16-01669-f021:**
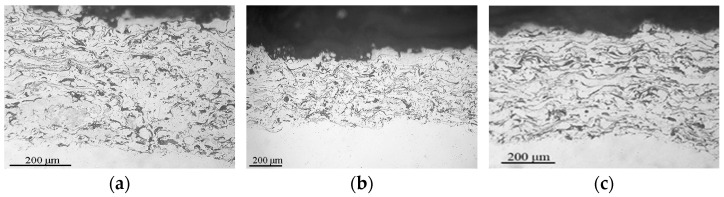
Microstructure of Fe-AlMg plasma coatings sprayed at (**a**) I = 500 A; (**b**) I = 450 A; (**c**) I = 500 A.

**Figure 22 materials-16-01669-f022:**
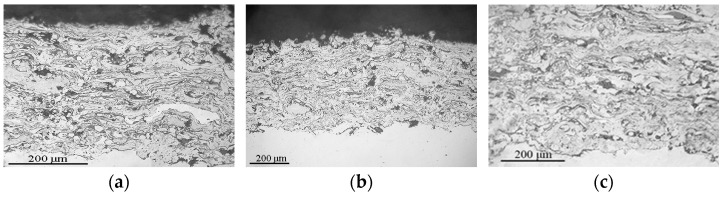
Microstructure of Fe-TiAl plasma coatings sprayed at (**a**) I = 500 A; (**b**) I = 450 A; (**c**) I = 500 A.

**Figure 23 materials-16-01669-f023:**
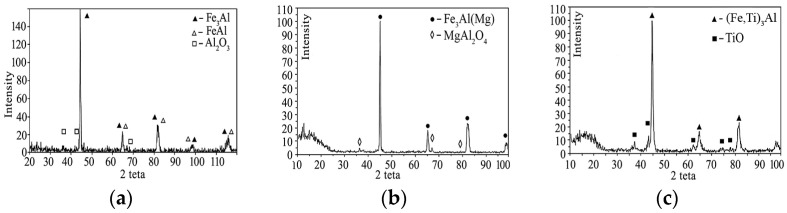
X-ray patterns of Fe_3_Al-based intermetallic plasma coatings sprayed at I = 500 A: (**a**) Fe_3_Al, (**b**) Fe-AlMg, (**c**) Fe-TiAl.

**Figure 24 materials-16-01669-f024:**
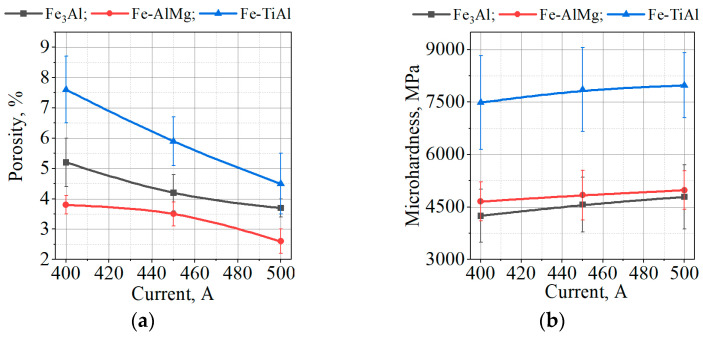
Effect of current change on (**a**) porosity and (**b**) microhardness of Fe_3_Al-based plasma coatings.

**Table 1 materials-16-01669-t001:** Chemical composition (wt.%) of MCS powders.

Spectrum	Fe	Al	Ti	Mg	O
Fe_3_Al Figure 1a					
1	79.22	16.91	-	-	3.87
2	81.78	14.04	-	-	4.18
3	81.35	13.51	-	-	5.14
4	80.15	14.79	-	-	5.06
Fe-AlMg Figure 1b					
1	82.14	13.99	-	0.62	3.25
2	82.49	12.39	-	0.55	4.57
Fe-TiAl Figure 1c					
1	61.72	12.03	21.21	-	5.04
2	59.43	13.59	22.17	-	4.81
3	60.97	11.14	24.24	-	3.65

**Table 2 materials-16-01669-t002:** Characteristics of Fe_3_Al–based intermetallic MCS powders.

System	Composition, wt.%	Phase Composition	Crystallite Size, nm	Microhardness, HV0.01 MPa	Particle Size, μm		
					D_10_	D_50_	D_90_
Fe_3_Al	86Fe+14Al	Fe_3_Al	15	4060 ± 1010	3.6	11.2	32.9
Fe-AlMg	86Fe+14(Al5Mg)	solid solution Mg in Fe_3_Al (Fe_3_Al(Mg))	14	4630 ± 950	2.8	14.5	29.8
Fe-TiAl	60.8Fe+39.2(Ti37.5Al)	solid solution Al in FeTi (Fe_1−x_TiAl_x_)	10	3400 ± 1120	2.6	8.7	29.7

**Table 3 materials-16-01669-t003:** Plasma spray modes for spraying of Fe_3_Al–based intermetallic particles.

Modes No.	Current, A	Voltage, V	Power, kW	PG Flow Rate, SLPM		Powder Feed Rate, g/min
				Ar	N_2_	
1	400	40	16	22	3	12
2	450		18			
3	500		20			

**Table 4 materials-16-01669-t004:** Chemical composition (wt.%) of Fe_3_Al powder particles after transfer through the plasma jet.

Figure	Spectrum	Fe	Al	O
Figure 6a	1	57.48	5.6	36.92
	2	84.31	9.96	5.73
	3	61.59	11.49	26.92
Figure 6b	1	76.83	10.08	13.09
	2	86.18	5.86	7.96
	3	36.36	30.7	32.94
Figure 6c	1	2.85	48.40	48.75
	2	82.97	6.62	10.41
	3	62.38	11.20	26.42
	4	76.40	10.48	13.12
	5	70.22	12.27	17.51

**Table 5 materials-16-01669-t005:** Chemical composition (wt.%) of Fe-AlMg powder particles after transfer through the plasma jet.

Figure	Spectrum	Fe	Al	Mg	O
Figure 7a	1	77.78	6.64	0.27	15.31
	2	71.58	5.76	0.52	22.14
	3	59.43	17.15	0.85	22.57
Figure 7b	1	9.56	48.93	0.30	41.21
	2	60.76	15.65	1.04	22.55
Figure 7c	1	61.89	11.2	0.81	26.1
	2	75.44	10.32	0.58	13.66
	3	52.52	10.55	0.3	36.63
	4	86.01	6.26	0.26	7.47

**Table 6 materials-16-01669-t006:** Chemical composition (wt.%) of Fe-TiAl powder particles after transfer through the plasma jet.

Figure	Spectrum	Fe	Al	Ti	O
Figure 8a	1	15.63	13.9	17.36	53.11
	2	20.59	12.92	17.78	48.71
	3	12.33	21.65	8.49	57.53
	4	16.41	14.32	15.6	53.67
Figure 8b	1	19.89	10.96	23.7	45.45
	2	3.87	6.0	45.85	44.28
	3	60.86	3.7	3.4	32.04
	4	24.5	9.09	19.4	47.01
Figure 8c	1	9.36	19.71	27.62	43.31
	2	19.16	11.92	32.08	36.84
	3	8.04	13.26	30.9	47.80
	4	6.32	7.46	40.47	45.75

**Table 7 materials-16-01669-t007:** Granulometric composition of powders after transfer through the plasma jet.

Powders	Current, A	Particle Size, μm		
		D_10_	D_50_	D_90_
Fe_3_Al	400	12	22	52
	450	13	25	52
	500	13	25	52
Fe-AlMg	400	13	19	40
	450	13	22	43
	500	13	28	58
Fe-TiAl	400	13	22	43
	450	13	25	46
	500	13	31	70

**Table 8 materials-16-01669-t008:** Degree of deformation of D/d particles after impact on the substrate.

Current, A	Powders	Average Particles Size d, μm	Average Splats Size D, μm	Particles Deformation Degree D/d
400	Fe_3_Al	22	46	2.1
	Fe-AlMg	19	49	2.6
	Fe-TiAl	22	42	1.9
450	Fe_3_Al	25	55	2.2
	Fe-AlMg	22	57	2.6
	Fe-TiAl	25	53	2.1
500	Fe_3_Al	25	55	2.2
	Fe-AlMg	28	78	2.8
	Fe-TiAl	31	65	2.1

**Table 9 materials-16-01669-t009:** Characteristics of Fe_3_Al-based intermetallic plasma sprayed coatings.

Coating	Current, A	Porosity, %	Microhardness, MPa	Phase Composition
Fe_3_Al	400	5.2 ± 0.8	4250 ± 760	Fe_3_Al, FeAl, traces Al_2_O_3_
	450	4.2 ± 0.6	4570 ± 780	
	500	3.7 ± 0.3	4790 ± 920	
Fe-AlMg	400	3.8 ± 0.3	4660 ± 560	Solid solution Mg in Fe_3_Al, MgAl_2_O_4_
	450	3.5 ± 0.4	4840 ± 710	
	500	2.6 ± 0.4	4980 ± 550	
Fe-TiAl	400	7.6 ± 1.1	7490 ± 1340	(Fe, Ti)_3_Al, TiO
	450	5.9 ± 0.8	7860 ± 1200	
	500	4.5 ± 1.0	7980 ± 930	

## Data Availability

The data are available in a publicly accessible repository.

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
