# Peer review of "Effect of APS Spraying Parameters on the Microstructure Formation of Fe3Al Intermetallics Coatings Using Mechanochemically Synthesized Nanocrystalline Fe-Al Powders"

_materials, 2023, doi:10.3390/ma16041669_

Round 1

Reviewer 1 Report

The authors presented the their flow of investigations in a good way, but latest related papers during 2022 may be included in literature part. 
The authors may justify why they have selected  the parameter variations specified by them with additional references. 

Author Response

Detailed Response to Reviewer’s Comments

Ms. Ref. No.: materials-2178115

Title: Effect of APS spraying parameters on the microstructure formation of Fe3Al intermetallics coatings using mechanochemically synthesized nanocrystalline Fe-Al powders Dear Sir or Madame,  

I would like to thank you very much for your letter and the reviewer’s comments on our manuscript (No.: materials-2178115). We appreciate your very valuable comments, which gave us a chance for revising the manuscript.

We have addressed all of the comments and revised the manuscript accordingly. All of the changes have been highlighted in yellow in the revised manuscript. Detailed responses to the comments are described in the “Response to Reviewers” point by point.

We now resubmit the manuscript for your further consideration for publication in your journal. We sincerely hope this revised manuscript will be finally accepted for publication. If you have any questions about this manuscript, please do not hesitate to contact me.

Best regards

Cezary Senderowski

On behalf of all co-authors

Institute of Mechanics and Printing

Warsaw University of Technology

E-mail: cezary.senderowski@pw.edu.pl

Reviewer’s Comments

We would like to thank the reviewers for their comments. The article was significantly improved according to all reviewers' comments and changes were highlighted.

Reviewer #1

  1. The authors presented the their flow of investigations in a good way, but latest related papers during 2022 may be included in literature part.

Response: Added reference [16] about Fe3Al-based powders for flame spraying of coatings (2022) to section 1. “Introduction”

  1. The authors may justify why they have selected the parameter variations specified by them with additional references.

Response: Added Fig. 3, 4 (the graphical result of simulation by the CASPSP software of plasma spraying of Fe3Al particles) to section 2. “Materials and Methods”

Reviewer 2 Report

The article " Effect of APS spraying parameters on the microstructure formation of Fe3Al intermetallics coatings using mechanochemically synthesized nanocrystalline Fe-Al powders" can be published after major revision.

The main weak point of the article is that the authors do not study the properties of the obtained powders and coatings (hardness, adhesion, corrosion resistance). This would improve the quality of the article.

In addition, the authors in the title state the synthesis of Fe-Al nanocrystal powders. However, this point is not proved in any way in the article. To do this, it is necessary to provide XRD analysis data for the synthesized powders

Author Response

Detailed Response to Reviewer’s Comments

Ms. Ref. No.: materials-2178115

Title: Effect of APS spraying parameters on the microstructure formation of Fe3Al intermetallics coatings using mechanochemically synthesized nanocrystalline Fe-Al powders Dear Sir or Madame,  

I would like to thank you very much for your letter and the reviewer’s comments on our manuscript (No.: materials-2178115). We appreciate your very valuable comments, which gave us a chance for revising the manuscript.

We have addressed all of the comments and revised the manuscript accordingly. All of the changes have been highlighted in yellow in the revised manuscript. Detailed responses to the comments are described in the “Response to Reviewers” point by point.

We now resubmit the manuscript for your further consideration for publication in your journal. We sincerely hope this revised manuscript will be finally accepted for publication. If you have any questions about this manuscript, please do not hesitate to contact me.

Best regards

Cezary Senderowski

On behalf of all co-authors

Institute of Mechanics and Printing

Warsaw University of Technology

E-mail: cezary.senderowski@pw.edu.pl

Reviewer’s Comments

We would like to thank the reviewers for their comments. The article was significantly improved according to all reviewers' comments and changes were highlighted.

Reviewer #2

The article "Effect of APS spraying parameters on the microstructure formation of Fe3Al intermetallics coatings using mechanochemically synthesized nanocrystalline Fe-Al powders" can be published after major revision.

We would like to thank the reviewer for the opinion and comments presented below.

  1. The main weak point of the article is that the authors do not study the properties of the obtained powders and coatings (hardness, adhesion, corrosion resistance). This would improve the quality of the article.

Response: Regarding your recommendation, the hardness results of particles were added to Table 2, as well as the porosity and hardness of coatings were added to Table 9. We also agree with the Reviewer's comment that the presentation of more widespread knowledge of some properties of the coatings is important for new information in the potential application of the Fe-Al type intermetallic coatings where important adhesion and corrosion resistance. These results we planning published future in other publications after a detailed investigation.

  1. In addition, the authors in the title state the synthesis of Fe-Al nanocrystal powders. However, this point is not proved in any way in the article. To do this, it is necessary to provide XRD analysis data for the synthesized powders.

Response: We would like to thank the reviewer for the comment which highly appreciate. The XRD patterns were detailed determine and provided by authors according to reviewer suggestions. The XRD results and analysis for the synthesized powders were added to section 2. "Materials and Methods".

Reviewer 3 Report

The manuscript entitled "Effect of APS spraying parameters on the microstructure formation of Fe3Al intermetallics coatings using mechanochemically synthesized nanocrystalline Fe-Al powders" presents the results of a study of the effect of atmospheric plasma spraying parameters on the properties of the applied powder. This topic is interesting and worthy to be investigated. However, this paper reports just simple characterization results without any scientific discussion.

There are a significant number of evident problems with the paper, which make it unacceptable for publication in the current form. The comments are listed below.

1. It is not clear why iron aluminide powders are called nanocrystalline.

2. The materials used are not characterized in terms of their chemical and phase composition.

3. For what reason for the synthesis of powders based on iron aluminides, in addition to pure aluminum powder, powders of aluminum alloy Al5Mg and Ti35Al intermetallic. Why these particular alloys? In Table 1, Ti35Al is designated as Ti37.5Al.

4. There is no information about the process of mechanochemical synthesis. How was it carried out?

5. EDS results are given only for powders obtained at a 500 A current value and only for Fe3Al coating.

6. There are no data on the structure of powders and coatings obtained at 400 and 450 A.

7. Data on the determination of the adhesive properties of coatings are not provided.

8. There are no results of characterization of the phase and chemical composition of plasma sprayed coatings.

9. Conclusions about the optimal modes of coating deposition are not substantiated. The choice of a specific ratio of the gas mixture is not at all clear.

Author Response

Detailed Response to Reviewer’s Comments

Ms. Ref. No.: materials-2178115

Title: Effect of APS spraying parameters on the microstructure formation of Fe3Al intermetallics coatings using mechanochemically synthesized nanocrystalline Fe-Al powders Dear Sir or Madame,  

I would like to thank you very much for your letter and the reviewer’s comments on our manuscript (No.: materials-2178115). We appreciate your very valuable comments, which gave us a chance for revising the manuscript.

We have addressed all of the comments and revised the manuscript accordingly. All of the changes have been highlighted in yellow in the revised manuscript. Detailed responses to the comments are described in the “Response to Reviewers” point by point.

We now resubmit the manuscript for your further consideration for publication in your journal. We sincerely hope this revised manuscript will be finally accepted for publication. If you have any questions about this manuscript, please do not hesitate to contact me.

Best regards

Cezary Senderowski

On behalf of all co-authors

Institute of Mechanics and Printing

Warsaw University of Technology

E-mail: cezary.senderowski@pw.edu.pl

Reviewer’s Comments

We would like to thank the reviewers for their comments. The article was significantly improved according to all reviewers' comments and changes were highlighted.

Reviewer #3

The manuscript entitled "Effect of APS spraying parameters on the microstructure formation of Fe3Al intermetallics coatings using mechanochemically synthesized nanocrystalline Fe-Al powders" presents the results of a study of the effect of atmospheric plasma spraying parameters on the properties of the applied powder. This topic is interesting and worthy to be investigated. However, this paper reports just simple characterization results without any scientific discussion.

There are a significant number of evident problems with the paper, which make it unacceptable for publication in the current form. The comments are listed below.

We would like to thank the reviewer for your opinion and all your comments.

  1. It is not clear why iron aluminide powders are called nanocrystalline.

Response: I would like to thank the reviewer for the comment which highly appreciates. In the revised description, based on XRD patterns added to section 2. "Materials and Methods", presented methodology and discussion we determined the nanocrystalline character of the iron aluminide powders.

  1. The materials used are not characterized in terms of their chemical and phase composition.

Response: In the submitted revision we provide a discussion of chemical and phase composition based on the SEM/EDS and XRD results added as Fig. 1, 2 and table 1 to section 2. "Materials and Methods"

  1. For what reason for the synthesis of powders based on iron aluminides, in addition to pure aluminum powder, powders of aluminum alloy Al5Mg and Ti35Al intermetallic. Why these particular alloys? In Table 1, Ti35Al is designated as Ti37.5Al.

Response: Regarding your question, the revised manuscript (section 2. Materials and Methods) made an additional discussion that " Using Ti as an alloying element allows realizing several mechanisms of iron intermetallic strengthening, namely structure ordering, strengthening with dispersed inclusions, formation of coherent microstructures. Titanium differs by significant solubility in solid state in Fe–Al phases that result in Fe3Al structure stabilizing about FeAl structure at high temperatures. Strengthening with dispersed precipitations of hexagonal Laves phase (Fe, Al)2Ti can take place in addition to strengthening due to structure ordering in Fe–Al–Ti system. Besides, there is a specific range of composition in the Fe–Al–Ti system where coherent structures [17] are formed. If an element such as Mg is used for alloying powders based on the Fe3Al intermetallic compound, it is possible to expect strengthening by incoherent compounds. Commercially available powders of alloys Al5Mg and Ti37.5Al were used instead of the pure elements of aluminium, magnesium and titanium to reduce the degree of oxidation of the MCS products.” Also, Ti35Al was changed to Ti37.5Al.

  1. There is no information about the process of mechanochemical synthesis. How was it carried out?

Response: Regarding your question, the revised manuscript (also section 2. Materials and Methods) made information about the process of mechanochemical synthesis "The MCS process was performed in a planetary-type mill «Aktivator 2SL» for 5 hours. The relation of the mass of balls to the mass of powder made 10:1. The central axis of the mill triboreactor was rotated at 100 rpm rate, drums rotated around their axis at 1500 rpm rate. Parts of the jar and milling agents were manufactured of 100Cr6 steel. The MCS process was performed in the air. Surface-active substances (SAS), namely oleic acid were added to the mixture to prevent pickup of processed charge on the milling agents and jar wall as well as the intensification of the process of new phases synthesis. The amount of aluminium alloy powder, introduced in the mixture with iron powder was selected for formation in MCS of Fe3(Al, Mg) intermetallics in the case of AlMg that corresponds to 14 wt.% of Al-alloy and (Fe, Ti)3Al in the case of TiAl intermetallic. In the latter variant amount of introduced TiAl made 39.2 wt.%. The microstructure, chemical and phase composition of the MCS powder products (as an intermetallic compound based on Fe3Al iron aluminide) were confirmed, respectively by SEM/EDS and XRD analysis using the JEOL 5310 microscope (Japan) operating at 20 kV, and XRD Simens D-500 diffractometer (Germany) with CoKα radiation (l= 0.178897 nm). An angular step size of 0.02o/min and a step time of 5 s per point were used respectively.”

  1. EDS results are given only for powders obtained at a 500 A current value and only for Fe3Al coating.

Response: In the submitted revision we provide both the EDS results (added in tables 4-6) and extended discussion about the other powders obtained at a 500 A current value, respectively Fe3Al, Fe-AlMg, and Fe-TiAl.

  1. There are no data on the structure of powders and coatings obtained at 400 and 450 A.

Response: In the submitted revision, the microstructure of powders and coatings obtained at 400 and 450 A was added in figures 9-11 and 20-22, respectively.

  1. Data on the determination of the adhesive properties of coatings are not provided.

Response: The adhesive of the coatings wasn't studied in this work. Such investigation will be performed in the future. Although, in figures 20 to 22 we observed the coherent structure of the coating volumes (in grain boundaries) as well as the Fe3Al coatings/substrate joint area. This fact underlies the claim that the splat boundaries and coatings/substrate joint area is not prone to the development of cracks and have a high degree of cohesive and adhesive strength.

  1. There are no results of characterization of the phase and chemical composition of plasma sprayed coatings.

Response: In the revised manuscript, the phase composition of plasma sprayed coatings are added to fig. 23 and table 9.

  1. Conclusions about the optimal modes of coating deposition are not substantiated. The choice of a specific ratio of the gas mixture is not at all clear.

Response: In the revised manuscript, based on the additional discussion it was found that using such parameters of plasma spray (voltage of 40 V, current of 500 A, plasma – Ar/N2 gas mixture in the ratio of 7.3:1) provides the formation of coating with more dense structure and higher microhardness. The Ar/N2 ratio (7.3/1) and plasma gas mixture flow rate (25 SLPM) make it possible to ensure stable operation of the plasmatron at a voltage of 40V. This information was added to section 2. “Materials and Methods.”

Round 2

Reviewer 2 Report

The paper can be accepted without any further changes.

Author Response

Dear Reviewer, 

We are grateful for your valuable comments, which allowed us to improve the quality of the manuscript approved by you for publication.
We sincerely thank you for finally being accepted for publication of this paper.

Sincerely

Cezary Senderowski

Reviewer 3 Report

The authors corrected the manuscript according to the reviewer's comments.

Please consider these suggestions:

1. Make EDS analysis points more visible (Fig. 1, 6, 7, 8).

2. Check dots/commas in values in tables 1, 5, 8.

3. Figures 3 and 4 are illegible. It would be possible to leave one figure to illustrate the capabilities of the software.

4. Microstructures of the coatings (Fig. 20, 21, and 22) should be the same size.

In general, subject to minor revisions, paper could be suitable for publication. 

Author Response

Dear Reviewer,

We are grateful for your valuable comments, which allowed us to improve the quality of the manuscript and pre-approve it for publication. We sincerely hope this revised manuscript will be finally accepted for publication.

The English language style and spell-checking have been improved according to your comments, and all changes have been highlighted in yellow in the revised manuscript. We have followed almost all of the reviewer's comments except point 3 "Figures 3 and 4 are illegible. It would be possible to leave one figure to illustrate the capabilities of the software."

According to the authors, Figures 3 and 4 clearly show the effect of changing the plasma spray mode on particle heating. In particular, the aggregation state of the particles is visible in these figures. Therefore, to justify the choice of spray modes, from our point of view, it is more to leave these drawings in the article.

Best regards

Cezary Senderowski

On behalf of all co-authors

Institute of Mechanics and Printing

Warsaw University of Technology

E-mail: cezary.senderowski@pw.edu.pl